

# Topological interpretation of color exchange invariants: Hexagonal lattice on a torus

**Olivier Cépas[1*] and Peter M. Akhmetiev[2,3]**

**1** Université Grenoble Alpes, CNRS, Grenoble INP, Institut Néel, 38000 Grenoble, France
**2** HSE Tikhonov Moscow Institute of Electronics and Mathematics,
34 Tallinskaya Str., 123458, Moscow, Russia
**3** Pushkov Institute of Terrestrial Magnetism, Ionosphere and Radio Wave Propagation,
Kaluzhskoe Hwy 4, 108840, Moscow, Troitsk, Russia

⋆ olivier.cepas@neel.cnrs.fr

## Abstract

We explain a correspondence between some invariants in the dynamics of color exchange in the coloring problem of a 2d regular hexagonal lattice, which are polynomials of winding numbers, and linking numbers in 3d. One invariant is visualized as linking of lines on a special surface with Arf-Kervaire invariant one, and is interpreted as resulting from an obstruction to transform the surface into its chiral image with special continuous deformations. We also consider additional constraints on the dynamics and see how the surface is modified.



# 1   Introduction

The use of color exchange was introduced a long time ago by Kempe in his attempt to prove that four colors were always sufficient to color a planar graph so that no two neighboring sites have the same color [1]. Color exchange consists of swapping colors in some regions of the graph -called "Kempe chains"-, in order to resolve conflicting sites, or, by extension, to create a new coloring. It was latter realized that such transformations cannot be used in general to reach *all* possible colorings of a graph: colorings form equivalence classes and there are obstructions to connect them [2].

The same issue appeared, more recently, in the physics of "frustrated" magnetism where dynamics of colors (that of spins) may consist precisely of successive color exchange transformations [3–5], rather than simple "spin-flips". The existence of classes of states, or "Kempe sectors", makes that physical dynamics nonergodic. It is a question to understand the obstructions and, possibly, to compute the invariants of the dynamics, which are characteristic of each class. The subject is at the frontier of discrete mathematics which have studied under which conditions the "reconfiguration graph" is connected [6,7], and statistical physics: *e.g.* coloring satisfaction problems, $q$-state Potts models, and gauge theories. In all these cases, the ergodicity issues are particularly important, not only for technical reasons, as in Monte Carlo sampling [3,4], but also in relation with glassy dynamics and conservation laws.

While these issues are, in the most general case, addressed for random graphs, it is also interesting to consider special graphs where they can be studied in details. In the magnetism of solids for example, *regular* lattices are most often relevant[1]. A well-known example is the three-color model on the hexagonal (honeycomb) lattice: it consists of coloring the edges with three colors in such a way that, around each vertex, the three incident edges have different colors (local constraints), see Fig. 1. Each proper "3-coloring" is a possible state for the model. Originally introduced as an exactly-solvable ice-type model, [9] it was argued to describe arrays of superconducting Josephson junctions [10], or some antiferromagnetic materials, the three colors being three spins at $120^o$, a configuration that minimizes the free-energy [3,4]. A natural dynamics can be defined within the 3-colorings, by allowing collective color exchanges along "Kempe chains". When the lattice has periodic boundary conditions, the Kempe chains take the form of closed loops of alternating colors. A remarkable fact is that, even when *all* loops of any size are included, the color exchange dynamics is nonergodic [3–5, 11]. This absence of ergodicity is related to the general issue mentioned above, but on the simpler setting of a special nonplanar graph which, because of the periodic boundary conditions, has the geometry of a "torus". In similarly constrained dimer models [12], the "transition graph" argument generally assures that the dynamics is ergodic when all loops are included. However, there are examples of dimer models where the motion of *short* loops in three-dimensional lattices results in classes of configurations [13,14]. The ergodicity is therefore a general issue in constrained systems and worth studying.

---

[1]Note though, that there are known physical examples of large magnetic molecules with spherical shapes where similar issues are at play, see *e.g.* Ref. [8].

Concerning the three-color model on the hexagonal lattice, the nonergodicity can be partly explained by the existence of a first invariant of the dynamics [15, 16], which can be seen, mathematically, as the parity of the *degree* of a color map [17–19]. This first invariant allows to describe two (odd/even) disconnected sectors of states, which are both extensive with system size [20]. However, other sectors exist [20]. Since the loops themselves evolve under the dynamics, it is natural to consider them up to some deformations, *i.e.* up to *homotopies*, and see them as elements of the fundamental group of the surface on which the lattice is defined. For example, for a lattice on a torus, that we will consider hereafter, the loops are characterized by their winding numbers (also called "fluxes" -the loops can be seen as "flux" lines). Individual winding numbers are not conserved but we recently argued that a complete classification of the sectors can be obtained by additional stable invariants that are polynomials of the winding numbers [21]. This needs to put aside the vast majority of sectors that result from steric constraints [21].

This raises the following question which we address in the present paper. While, for example, standard vortices are characterized by their (integer) vorticity -or elements of the first homotopy group-, is it possible to provide a similar topological interpretation of the invariant polynomials of the winding numbers? Is there a known obstruction characterized by homotopy group elements? We will see that the *linking* numbers of the Kempe loops are the natural numbers to visualize some of these invariants. However, since linking numbers are defined in three, not in two, dimensions, it is necessary to visualize the surface on which the lattice is defined in three dimensions and there are several inequivalent ways to do it [22]. Note that, in the dynamical evolution, the loops can cross and be subject to "surgeries", with cutting and reconnections, so that there is no guarantee that linking numbers will be conserved in any way.

To study this question, we will broaden the color dynamics we consider, and introduce another special example (within a whole hierarchy of possible dynamics). The first dynamics is the one directly motivated by the physical color problem [21], which we call "dynamics I" here in section 2.1. This is not the only possibility and one can add other types of constraints and see whether or not these additional constraints generate nontrivial sectors: this is the object of "dynamics II" considered in section 2.2, where the additional constraint is a "parity control". Since adding more and more constraints further create imbricated sectors, this is part of a whole hierarchy of dynamics. Formally, it can be seen as a process to forbid, in the space of configurations, certain types of "moderate singularities", higher than a given type [23, 24][2]: disconnected and imbricated classes may form as a result.

We will give the invariants of both dynamics in section 2. In section 3.1, we will see that a color invariant of "dynamics II" is directly interpreted as linking numbers on the *standard torus in 3d*. This surface has to be modified for "dynamics I" in terms of special surfaces with self-intersections, *i.e. immersions* in 3d. We will explain how to define the linking numbers, first on a simple instance of immersion in section 3.2, and, second, on general immersions in section 3.3. In what sense such linking numbers are topological invariants of surfaces is explained in section 3.4. In section 4, we discuss the two equivalence classes of torus immersions, and, in section 5 why we need a special immersion to interpret the invariant of "dynamics I", and why the color invariant reflects the obstruction to transform the corresponding immersion into its chiral image.

---

[2]More precisely, in the sense of Refs. [23, 24], the complements of discriminants may form disconnected classes when some "moderate singularities" are forbidden to occur.

## 2 Invariants of 2d flux lines dynamics

### 2.1 Dynamics I

The model is defined as a color satisfaction problem which consists of coloring the edges of the 2d hexagonal lattice with three colors A, B, and C such that no two neighboring edges have the same color (see Fig. 1 for an example). There is an exponential number of such proper 3-colored states with system size [9].

The dynamics within this ensemble of states is simply defined by exchanging colors in a particular way. For this, three types of loops are defined: they are made of successive edges of two alternating colors A-B ("$c$-type"), A-C ("$b$-type") and B-C ("$a$-type"), called "Kempe chains" along which the two colors can be exchanged. On a lattice with periodic boundary conditions, as is considered here (*i.e.* the lattice is defined on a torus $T^2$), such a loop is always closed and of even length. All three types of loops, of *all* lengths, can be "flipped"; this defines the "dynamics" of the model.

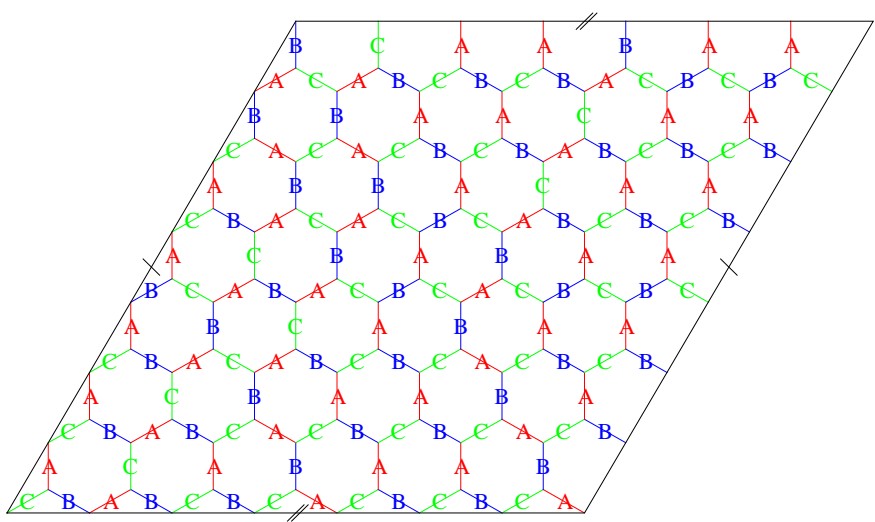

Figure 1: Example of a 3-coloring of a hexagonal lattice, with periodic boundary conditions (torus geometry).

Any 2-colored loop can be oriented (see Fig. 2 for the definition of the orientation) and is characterized by two integer winding numbers, or an integer vector, $\boldsymbol{w} = (p, q)$ that give how many times it winds around the torus. It can be seen, up to homotopy, as an element of the fundamental group of the torus, $\pi_1(T^2) = \mathbb{Z} \times \mathbb{Z}$. Considering all loops, a given color configuration is thus characterized by a triplet of integer vectors,

$$(\boldsymbol{a}, \boldsymbol{b}, \boldsymbol{c}), \tag{1}$$

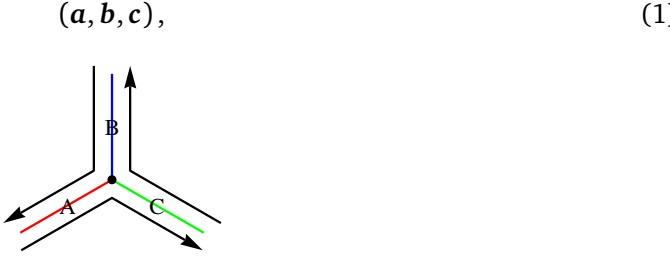

Figure 2: Definition of the orientation of the loops for a black vertex (the definition is opposite on white vertices).

where $\boldsymbol{a} = (a_x, a_y)$ defines the two *total* integer winding numbers (or "fluxes") of all B-C loops, $\boldsymbol{b} = (b_x, b_y)$ defines the two winding numbers of all C-A loops and, similarly, $\boldsymbol{c} = (c_x, c_y)$ for the A-B loops (we will sometimes use a third component $a_z$ defined by $a_z \equiv -a_x - a_y$). They are not all independent in the present coloring problem: the triplet of vectors $(\boldsymbol{a}, \boldsymbol{b}, \boldsymbol{c})$ must satisfy the equation [21],

$$\boldsymbol{a} + \boldsymbol{b} + \boldsymbol{c} = 0, \tag{2}$$

showing that there are four independent integers, *e.g.* $(a_x, a_y, b_x, b_y)$.

Color exchange of the two colors of any loop, and more particularly of winding loops, leads to a transformation of $(\boldsymbol{a}, \boldsymbol{b}, \boldsymbol{c})$. The evolution of the triplet of integer vectors $(\boldsymbol{a}, \boldsymbol{b}, \boldsymbol{c})$ thus defines an integer dynamics, governed by special transformations. The three possible transformations from $(\boldsymbol{a}, \boldsymbol{b}, \boldsymbol{c})$ to $(\boldsymbol{a}', \boldsymbol{b}', \boldsymbol{c}')$ are given as follows [21], depending on which $a$, $b$, or $c$-type loop flips,

$$(\boldsymbol{a}', \boldsymbol{b}', \boldsymbol{c}') = (\boldsymbol{a} + 2k\hat{\boldsymbol{a}}, \boldsymbol{b} - k\hat{\boldsymbol{a}}, \boldsymbol{c} - k\hat{\boldsymbol{a}}), \tag{3}$$

$$(\boldsymbol{a}', \boldsymbol{b}', \boldsymbol{c}') = (\boldsymbol{a} - k\hat{\boldsymbol{b}}, \boldsymbol{b} + 2k\hat{\boldsymbol{b}}, \boldsymbol{c} - k\hat{\boldsymbol{b}}), \tag{4}$$

$$(\boldsymbol{a}', \boldsymbol{b}', \boldsymbol{c}') = (\boldsymbol{a} - k\hat{\boldsymbol{c}}, \boldsymbol{b} - k\hat{\boldsymbol{c}}, \boldsymbol{c} + 2k\hat{\boldsymbol{c}}), \tag{5}$$

where $\hat{\boldsymbol{a}} = \frac{\boldsymbol{a}}{\gcd(a_x, a_y)}$ is a *primitive* vector along $\boldsymbol{a}$, $\gcd(a_x, a_y)$ is the greatest common divisor of $a_x$ and $a_y$ and is basically the number of disconnected winding loops of type $a$. The positive or negative integer $k$ gives the possibility to insert pairs of parallel winding loops or flip them. Since $\boldsymbol{a}$, $\boldsymbol{b}$, $\boldsymbol{c}$ play the same role, it is possible to permute them (up to a change of sign) by a suitable choice of $k$. The integer dynamics described by Eqs.(3)-(5), called dynamics I, is that of the original coloring problem, coarse-grained to the level of winding numbers.

The dynamics has several constraints [21]. The first constraint is a steric constraint; the lattice has finite linear size $L$ and there is a maximal packing of winding loops, giving necessarily,

$$|a_\alpha| \le L, \qquad |b_\alpha| \le L, \qquad |c_\alpha| \le L, \tag{6}$$

where $\alpha = x, y$ or $z$. The four independent integers therefore evolve inside a "box", $[-L, L]^4$. The second constraint arises from the interpretation of loops as colors, and takes the form of linear constraints,

$$\boldsymbol{a} - \boldsymbol{b} = \boldsymbol{L} \pmod 3, \tag{7}$$

$$\boldsymbol{c} - \boldsymbol{a} = \boldsymbol{L} \pmod 3, \tag{8}$$

$$\boldsymbol{b} - \boldsymbol{c} = \boldsymbol{L} \pmod 3, \tag{9}$$

where $\boldsymbol{L}$ is a notation for $(L, L)$. Not all sets of four independent integers $(a_x, b_x, a_y, b_y)$ are therefore allowed and the constraints can be seen as partitioning the box in sectors characterized by $L \pmod 3$. Such constraints *e.g.* $\boldsymbol{a}' - \boldsymbol{b}' = \boldsymbol{a} - \boldsymbol{b} + 3k\hat{\boldsymbol{a}}$, are naturally conserved, modulo 3, by the dynamics. What is remarkable is that these sectors split into subsectors. The problem is to classify equivalent classes of triplets $(\boldsymbol{a}, \boldsymbol{b}, \boldsymbol{c})$ up to the transformations given by Eqs. (3)-(5).

We have several invariants [21], which we recall,

$$I_0 = \boldsymbol{a} + \boldsymbol{b} + \boldsymbol{c}, \tag{10}$$

$$I_1 = \frac{1}{3}(\boldsymbol{a} \times \boldsymbol{b} + \boldsymbol{b} \times \boldsymbol{c} + \boldsymbol{c} \times \boldsymbol{a}) \pmod 2. \tag{11}$$

$I_0 = 0$ for the physical color problem. The case $I_1 = 0$ corresponds to a connected "even" sector. The case $I_1 = 1$ corresponds to the "odd" sectors, which have two additional invariants,

$$I_2 = \frac{1}{3}(\boldsymbol{a}.\boldsymbol{b} + \boldsymbol{b}.\boldsymbol{c} + \boldsymbol{c}.\boldsymbol{a}) \pmod 4, \tag{12}$$

$$I_3^\pm = P_5^\pm(\boldsymbol{a}, \boldsymbol{b}, \boldsymbol{c}) \pmod 4, \tag{13}$$

where $I_2 = \pm 1$ and $I_3^{\pm} = \pm 1$ (meant modulo 4) and the upper script $\pm$ means for the corresponding value of $I_2 = \pm 1$. $P_5^{\pm}(\boldsymbol{a}, \boldsymbol{b}, \boldsymbol{c})$ is a polynomial of degree 5 (symmetrical under the permutations of $\boldsymbol{a}$, $\boldsymbol{b}$ and $\boldsymbol{c}$) [21]. We thus have 5 sectors: a connected sector with $I_1 = 0$ (even sector) and 4 sectors for $I_1 = 1$ (odd sectors), fully described by $I_2 = \pm 1$ and $I_3^{\pm} = \pm 1$. Additional sectors exist, but are "unstable" when the "box" constraints given by Eq. 6 are relaxed to $L_1 \geq L$. These sectors get reconnected through higher winding number configurations and are thus not related to a stable invariant. The set $I_0, I_1, I_2$ and $I_3^{\pm}$ was argued to be a complete set of stable invariants for dynamics I [21].

While $I_1$ has a simple interpretation in terms of the parity of the number of signed crossings of winding loops, we will be interested here in interpreting $I_2$, and leave the interpretation of $I_3^{\pm}$ as an open question.

Since by permutation, it is always possible to consider configurations with a given parity of the winding numbers, $(\boldsymbol{a}, \boldsymbol{b}, \boldsymbol{c})$ (mod 2), for example $(1, 0, 1, 0, 1, 1, 1, 1, 0)$ (one of the six possibilities in an odd sector), we can rewrite $I_2$ as

$$I_2 = 2(a_x b_x + a_y b_y) + a_x b_y + a_y b_x \quad (\text{mod } 4). \tag{14}$$

This is under this form that we will interpret $I_2$ as a linking number of $a$ and $b$ curves on a special immersion (section 5).

## 2.2 Dynamics II

We now include an additional constraint, a "parity control" to the previous dynamics and call it "dynamics II". This choice is not directly motivated by the original physical color problem but the addition of constraints can lead to further nontrivial sectors and is interesting as a classification problem. Moreover, it is not impossible that new invariants could help to classify the unstable sectors of the original dynamics [21] (it is possible, in principle, that an additional constraint mimics the size constraint of some unstable sectors), a point not investigated here. Here, we will show numerically that the sectors split into 23 stable sectors (instead of 16 if there were no new nontrivial sectors as we will see below) and will give the invariants explicitly.

The additional constraints we consider are

$$
\begin{align}
a_x' &= a_x \quad (\text{mod } 2), \tag{15} \\
b_x' &= b_x \quad (\text{mod } 2), \tag{16} \\
c_x' &= c_x \quad (\text{mod } 2), \tag{17}
\end{align}
$$

where $a_x'$ is the $x$-component of $\boldsymbol{a}'$ (after a color exchange). We thus impose a "parity control" on

$$R = (a_x, b_x, c_x) \quad (\text{mod } 2), \tag{18}$$

which the dynamics II must conserve: the loops can be flipped only if the parity of their winding numbers in the $x-$direction is conserved.

The previous invariants still hold, since the dynamical equations are the same and we can consider, in particular, even sectors ($I_1 = 0$) and odd sectors ($I_1 = 1$).

We have numerically iterated the dynamics, $(\boldsymbol{a}, \boldsymbol{b}, \boldsymbol{c}) \rightarrow (\boldsymbol{a}', \boldsymbol{b}', \boldsymbol{c}')$ [Eqs. (3)-(5)] under all previous and current constraints, i.e. by rejecting the moves that do not satisfy the constraints. We use a numerical depth-first search algorithm, which terminates once all states connected to an initial state are exhausted and thus construct the sectors.

We find that the even sector ($I_1 = 0$), which was *connected* in dynamics I, now splits into many subsectors. In Fig. 3, we give the result of the distribution of sizes of each even Kempe sector up to $L = 100$. We find a very large number of small sectors (containing typically less

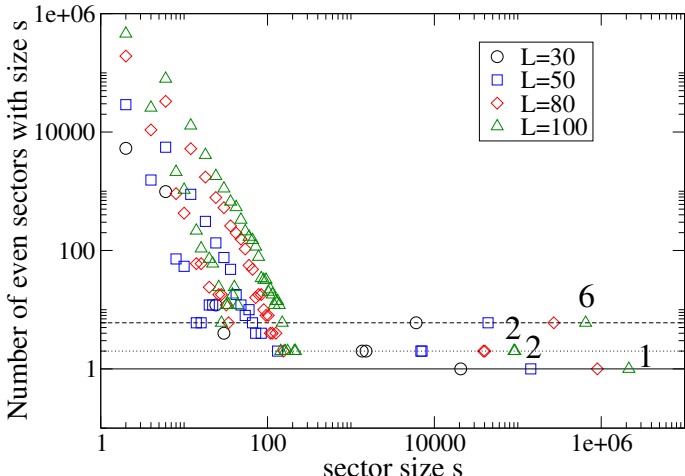

Figure 3: Distribution of sizes of even sectors of dynamics II for various linear sizes $L$ (logarithmic scales). We note a numerous number of small sectors and 11 large sectors, with a clear separation between the two.

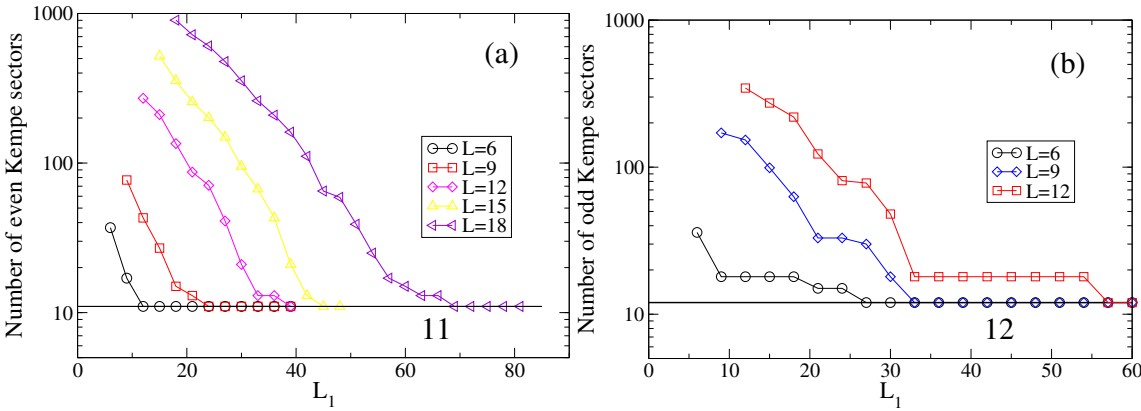

Figure 4: Stabilization of the number of even (a) and odd (b) sectors. The two numbers 11 and 12 give the number of stable sectors (invariant-related).

than $L$ configurations) and 11 large sectors. Some have symmetries as seen in the degeneracies: six sectors, on one hand, have the same size, and there are two doublet sectors).

In Fig. 4, we study the convergence of the number of sectors when the "box" constraints given by Eq. 6 are progressively relaxed by allowing intermediate configurations with winding numbers up to $L_1 \geq L$. It gives the convergence of the number of sectors with size: 11 even sectors and 12 odd sectors and reflects the reconnections of the smaller sectors, which, as we argued earlier [21], are not related to stable invariants.

In order to study the stable invariants, we generate only the configurations that belong to the large sectors, by introducing a size cutoff, typically a few $L$ (as found in Fig. 3). In this way, we can study the regularities in the winding numbers and extract the stable invariants.

We can first consider sectors according to their conserved parity $R$. Among the $2^3 = 8$ possibilities for $R$, we have only 4 since $a_x + b_x + c_x = 0$: $R = (0,0,0); (0,1,1); (1,0,1); (1,1,0)$.

In the even sector, all four possibilities,

$$R = (0,0,0); (0,1,1); (1,0,1); (1,1,0), \tag{19}$$

are compatible with even $\chi = a_x b_y - a_y b_x$, giving four trivial sectors. In fact, the sector

Table 1: The splitting of the even sector ($I_1 = 0$) into 11 subsectors conserved by dynamics II and classified by the imposed "parity control" $R$ and the nontrivial invariants $I'_2$, $I''_2$, $I_3^{\pm'}$, for $L = 0,1$ (mod 3) (for $L = 2$ (mod 3), change all signs in $(a,b,c)$ of $L = 1$ (mod 3)). Examples of configurations $(a,b,c)$, (with the smallest norm $n^2 = \frac{1}{6}\sum_{i=x,y,z}(a_i^2 + b_i^2 + c_i^2)$) are given.

|  | Sect. | $R$ | $I'_2$ | $I''_2$ | $I_3^{\pm'}$ | $(a,b,c)$ | $n^2$ |
|---|---|---|---|---|---|---|---|
| | 1 | (0,0,0) | 0 | - | - | 0 | 0 |
| | 2 | (0,0,0) | 1 | 1 | -1 | $(2,-2,0,2,1,-3,-4,1,3)$ | 8 |
| | 3 | (0,0,0) | 1 | 1 | 1 | $-(2,-2,0,2,1,-3,-4,1,3)$ | 8 |
| | 4 | (0,0,0) | 1 | -1 | -1 | $-(2,0,-2,2,-3,1,-4,3,1)$ | 8 |
| $L = 0$ (mod 3) | 5 | (0,0,0) | 1 | -1 | 1 | $(2,0,-2,2,-3,1,-4,3,1)$ | 8 |
| | 6 | (0,1,1) | 0 | - | - | $\pm(2,0,-2,-1,0,1,-1,0,1)$ | 2 |
| | 7 | (0,1,1) | 1 | - | - | $\pm(2,-2,0,-1,1,0,-1,1,0)$ | 2 |
| | 8 | (1,0,1) | 0 | - | - | $\pm(-1,0,1,2,0,-2,-1,0,1)$ | 2 |
| | 9 | (1,0,1) | 1 | - | - | $\pm(-1,1,0,2,-2,0,-1,1,0)$ | 2 |
| | 10 | (1,1,0) | 0 | - | - | $\pm(-1,0,1,-1,0,1,2,0,-2)$ | 2 |
| | 11 | (1,1,0) | 1 | - | - | $\pm(-1,1,0,-1,1,0,2,-2,0)$ | 2 |
|  | Sect. | $R$ | $I'_2$ | $I''_2$ | $I_3^{\pm'}$ | $(a,b,c)$ | $n^2$ |
| | 1 | (0,0,0) | 0 | - | - | $(0,0,0,2,-1,-1,-2,1,1)$ | 2 |
| | 2 | (0,0,0) | 1 | 1 | -1 | $(2,-2,0,4,-3,-1,-6,5,1)$ | 16 |
| | 3 | (0,0,0) | 1 | 1 | 1 | $(2,0,-2,-2,-1,3,0,1,-1)$ | 4 |
| | 4 | (0,0,0) | 1 | -1 | -1 | $(2,-2,0,-2,3,-1,0,-1,1)$ | 4 |
| $L = 1$ (mod 3) | 5 | (0,0,0) | 1 | -1 | 1 | $(2,0,-2,4,-1,-3,-6,1,5)$ | 16 |
| | 6 | (0,1,1) | 0 | - | - | $(0,0,0,-1,2,-1,1,-2,1)$ | 2 |
| | 7 | (0,1,1) | 1 | - | - | $(0,0,0,-1,-1,2,1,1,-2)$ | 2 |
| | 8 | (1,0,1) | 0 | - | - | $(-1,2,-1,0,0,0,1,-2,1)$ | 2 |
| | 9 | (1,0,1) | 1 | - | - | $(-1,-1,2,0,0,0,1,1,-2)$ | 2 |
| | 10 | (1,1,0) | 0 | - | - | $(-1,2,-1,1,-2,1,0,0,0)$ | 2 |
| | 11 | (1,1,0) | 1 | - | - | $(-1,-1,2,1,1,-2,0,0,0)$ | 2 |

$R = (0,0,0)$ splits into 5 sectors, see Table 1, and the three others split into 2 sectors, so that we get the 11 sectors of the even sector (as found in Fig. 3 and Fig. 4 (a)).

In the odd sector, there remains only three possibilities among four,

$$R = (0,1,1); (1,0,1); (1,1,0), \tag{20}$$

because $(0,0,0)$ is not compatible with an odd $\chi$. Since each of the three must be split into four, according to the invariants described in section 2.1, this simply explains the 12 odd sectors observed. It shows that there are no additional nontrivial sectors and invariants in the odd sector. We now describe the nontrivial sectors of the even sector.

### 2.2.1 Additional invariants of the even sector.

Among the four possibilities of parity control (19), consider first $R = (0,1,1)$, $(1,0,1)$ or $(1,1,0)$. We have found numerically that

$$I'_2 = \frac{a_x b_y + a_y b_x + b_x c_y + b_y c_x + c_x a_y + c_y a_x}{2} \quad (\text{mod } 2), \tag{21}$$

is invariant in the dynamics II (when $I_1 = 0$). In this expression, there are always four terms that cancel. For example, for $R = (0, 1, 1)$,

$$I_2' = \frac{b_x c_y + b_y c_x}{2} \quad (\text{mod } 2). \tag{22}$$

Indeed, since $I_1 = 0$, $a_x b_y - a_y b_x$ is even, $a_x$ is even so $a_x b_y$ is even. $b_x$ is odd so $a_y$ must be even. Since $a_x + a_y + a_z = 0$, one must have $a_z$ even. Therefore $\boldsymbol{a} \ (\text{mod } 2) = 0$. We see that $(a_x b_y + a_x c_y)/2 = a_x(b_y + c_y)/2 = -a_x a_y/2$ is an integer multiple of two. The same argument applies to $(a_y b_x + a_y c_x)/2$.

Second, consider $R = (0, 0, 0)$. Since, by definition, $a_x, b_x$ and $c_x$ are even, define,

$$\tilde{\boldsymbol{a}} = (a_x/2, a_y, -a_x/2 - a_y), \tag{23}$$

$$\tilde{\boldsymbol{b}} = (b_x/2, b_y, -b_x/2 - b_y), \tag{24}$$

$$\tilde{\boldsymbol{c}} = (c_x/2, c_y, -c_x/2 - c_y). \tag{25}$$

They are all integer numbers and all constraints are satisfied, $\tilde{\boldsymbol{a}} + \tilde{\boldsymbol{b}} + \tilde{\boldsymbol{c}} = 0$, $\tilde{a}_x + \tilde{a}_y + \tilde{a}_z = 0$ *etc*. We also have

$$\tilde{a}_x \tilde{b}_y - \tilde{a}_y \tilde{b}_x = \chi/2 = 1 + 2n = 1 \quad (\text{mod } 2), \tag{26}$$

so that $(\tilde{\boldsymbol{a}}, \tilde{\boldsymbol{b}}, \tilde{\boldsymbol{c}})$ is a valid state in an odd sector. We can therefore label it by $I_2$ and $I_3^{\pm}$ of section 2.1. We can apply the dynamics to $(\tilde{\boldsymbol{a}}, \tilde{\boldsymbol{b}}, \tilde{\boldsymbol{c}})$ and generate a new state $(\tilde{\boldsymbol{a}}', \tilde{\boldsymbol{b}}', \tilde{\boldsymbol{c}}')$. Then, by inverting the linear transformation above, we have a new state $(\boldsymbol{a}', \boldsymbol{b}', \boldsymbol{c}')$ which is in the same sector as $(\boldsymbol{a}, \boldsymbol{b}, \boldsymbol{c})$, labeled by the same invariants. Knowing the invariants for odd sectors [Eq. 12], we generate new invariants for the even sectors:

$$I_2'' = \frac{1}{3}(\tilde{\boldsymbol{a}}.\tilde{\boldsymbol{b}} + \tilde{\boldsymbol{b}}.\tilde{\boldsymbol{c}} + \tilde{\boldsymbol{c}}.\tilde{\boldsymbol{a}}) \quad (\text{mod } 4), \tag{27}$$

the three terms giving equal contribution. We replace and obtain

$$\tilde{\boldsymbol{a}}.\tilde{\boldsymbol{b}} = \frac{a_x b_x}{2} + 2 a_y b_y + \frac{1}{2}(a_x b_y + a_y b_x). \tag{28}$$

By noticing that,

$$\frac{1}{3}\left[\frac{1}{2}(a_x b_x + b_x c_x + c_x a_x) + 2(a_y b_y + b_y c_y + c_y a_y)\right], \tag{29}$$

vanishes modulo 4 in these sectors, $I_2''$ simplifies:

$$I_2'' = \frac{1}{6}(a_x b_y + a_y b_x + b_x c_y + b_y c_x + c_x a_y + c_y a_x) \quad (\text{mod } 4). \tag{30}$$

Similarly, we define, $I_3^{\pm'}(\boldsymbol{a}, \boldsymbol{b}, \boldsymbol{c}) = P_5^{\pm}(\tilde{\boldsymbol{a}}, \tilde{\boldsymbol{b}}, \tilde{\boldsymbol{c}}) \ (\text{mod } 4)$, where $P_5^{\pm}$ is the corresponding invariant of dynamics I.

The integer numbers $R$, $I_2'$, $I_2''$ and $I_3^{\pm'}$ give a complete classification of the 11 stable even sectors ($I_1 = 0$) of dynamics II, while $R$, $I_2$, and $I_3^{\pm}$ give a complete classification of the 12 stable odd sectors ($I_1 = 1$).

## 3 Definition of the linking number

The linking number of two curves in 3d space is defined, as usual, as the number of signed intersections of one curve with the surface whose boundary is the second curve, called a Seifert surface.

When two curves have intersection points, there is an ambiguity in the definition of the linking number. We generally average over all possible resolutions of the intersections, by shifting one of the two curves in the two possible ways away from the intersection point. This extends the linking numbers to half-integer values, the half-integer part representing the number of intersection points.

## 3.1 Linking number on the standard torus in 3d

The linking number for arbitrary curves on the embedded (with no self-intersection) torus in 3d is defined as a bilinear form of the cycles. It depends only on the homology class and not on the cycles themselves. Consider a cycle as a vector in a vector space ($H_1(T^2, \mathbb{Z})$, the first homology group with coefficients in $\mathbb{Z}$), which can be written

$$\boldsymbol{a} = a_x \boldsymbol{x} + a_y \boldsymbol{y}, \tag{31}$$

where $\boldsymbol{x}$ and $\boldsymbol{y}$ are two basis cycles and $a_x$, $a_y$ the two integer winding numbers. Then, the linking number of $\boldsymbol{a}$ and $\boldsymbol{b}$ is defined as

$$\text{lk}(\boldsymbol{a}, \boldsymbol{b}) = a_x b_x \text{lk}(\boldsymbol{x}, \boldsymbol{x}) + a_x b_y \text{lk}(\boldsymbol{x}, \boldsymbol{y}) + a_y b_x \text{lk}(\boldsymbol{y}, \boldsymbol{x}) + a_y b_y \text{lk}(\boldsymbol{y}, \boldsymbol{y}). \tag{32}$$

Note that $\text{lk}(\boldsymbol{x}, \boldsymbol{x}) = \text{lk}(\boldsymbol{y}, \boldsymbol{y}) = 0$ and $\text{lk}(\boldsymbol{x}, \boldsymbol{y}) = \text{lk}(\boldsymbol{y}, \boldsymbol{x}) = 1/2$, since the cycles $\boldsymbol{x}$ and $\boldsymbol{y}$ have one intersection point, so that

$$\text{lk}(\boldsymbol{a}, \boldsymbol{b}) = \frac{1}{2}(a_x b_y + a_y b_x). \tag{33}$$

For example, the curve that corresponds to the cycle $\boldsymbol{a} = (1, 1)$ has $\text{lk}(\boldsymbol{a}, \boldsymbol{a}) = 1$, it is "self-linked", or is a "torus knot".

This form, Eq. (33), of the linking number actually corresponds to the invariant $I_2'$ of the dynamics II, Eq. (21), and is, therefore, conserved by dynamics II. It appears now as obvious that sectors 6 and 7 for instance (see table 1), with, respectively, $I_2' = 0$ and $I_2' = 1$, can be seen as two unlinked (resp. linked), loops. In sector 6, the configuration with $\boldsymbol{b} = (1, 0)$ and $\boldsymbol{c} = (1, 0)$ are two parallel unlinked loops, $\text{lk}(\boldsymbol{b}, \boldsymbol{c}) = 0$. In sector 7, $\boldsymbol{b} = (1, -1)$ and $\boldsymbol{c} = (1, -1)$ are two parallel linked loops, $\text{lk}(\boldsymbol{b}, \boldsymbol{c}) = 1$. Since the invariant is conserved only modulo 2, there is no higher linking number. The obstruction characterized by $I_2'$ therefore simply corresponds to the impossibility to dynamically unlink the two loops.

In sectors with $R = (0, 0, 0)$, the conservation of $I_2''$ can also be expressed in terms of linking numbers. From Eq. (30),

$$I_2'' = \frac{1}{3}\left[\text{lk}(\boldsymbol{a}, \boldsymbol{b}) + \text{lk}(\boldsymbol{b}, \boldsymbol{c}) + \text{lk}(\boldsymbol{c}, \boldsymbol{a})\right] \pmod{4}, \tag{34}$$

and the same interpretation holds, modulo 4.

In order to describe other invariants such as $I_2$, it is necessary to modify the expression of the linking form, Eq. (33) and change the linking properties of the basis cycles, *e.g.* $\text{lk}(\boldsymbol{x}, \boldsymbol{x})$, in Eq. (32). As we will see, this results in surfaces with self-intersections in 3d, i.e. *immersions*.

## 3.2 Linking number on the double cover of the torus

We consider here the simplest example of an immersion of a torus to explain why, in general, the linking number on immersions is *ill-defined*. The linking number of two curves will indeed depend on the relative positions of them and when one of them is deformed (evolves dynamically), the linking number will change. To understand this point, consider the immersion shown in Fig. 5 (b). It is a double cover of the torus obtained by identifying the edges

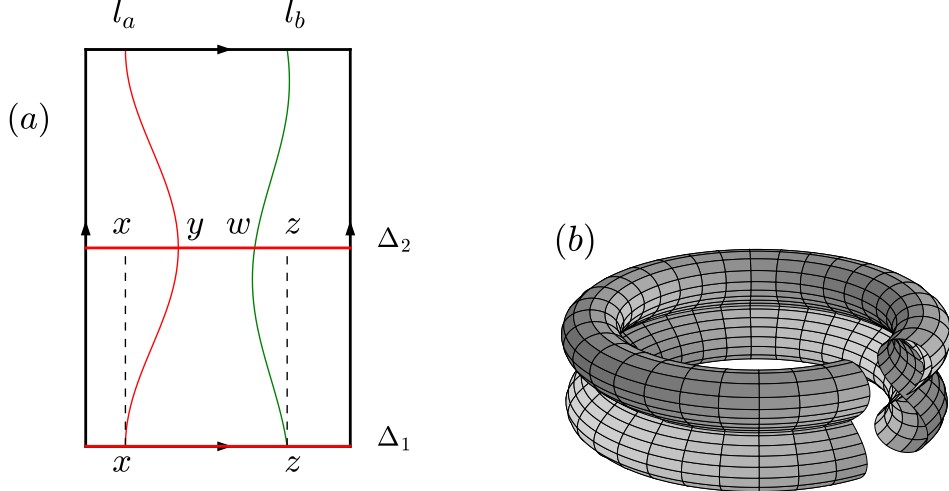

Figure 5: A surface with the topology of a torus, with the identification of $\Delta_1$ and $\Delta_2$ and the edges (a), which is a double cover of the torus over a parallel $\Delta_1$. Two arbitrary closed loops $l_a$ and $l_b$ are represented. An immersion of it in 3d is shown in (b), with its self-intersection curve $\Delta$, with preimages $\Delta_1$ and $\Delta_2$.

of the surface (Fig. 5 (a)) and mapping the two curves $\Delta_1$ and $\Delta_2$ on $\Delta$, which becomes a self-intersection curve in 3d.

Consider, in Fig. 5 (a), a first closed loop $l_a$ which cuts $\Delta_1$ in $x$ and $\Delta_2$ in $y$ and a second loop $l_b$ which cuts them in $w$ and $z$. When $l_a$ moves in its homotopy class, the point $y$ of $l_a$, in particular, will move on $\Delta_2$ and may cross the point $z$ of $l_b$: it gives an intersection point of the corresponding curves $L_a$ and $L_b$ on the immersion in 3d, and, hence, a jump of the linking number by $\pm 1$ (depending on the orientation).

We say that the couple of points $(y, z)$ are "ordered" if they occur in this order on $\Delta$ and introduce a number $\sharp(l_a, l_b) = 0$ in this case. If $y$ crosses over $z$, $(z, y)$ are "disordered" by a single permutation, so that we define $\sharp(l_a, l_b) = 1$ to compensate the jump of the linking number. We see also that moving homotopically $l_a$ to the left in Fig. 5 (a) and crossing the boundary will permute $y$ and $z$: the couples $(y, z)$ and $(z, y)$ are physically equivalent, or in other words, $\sharp(l_a, l_b)$ is defined only modulo 1.

In this case, the linking number itself is defined only modulo 1. Since it can be a half-integer or an integer, it does not reflect anything else than the parity of crossings.

## 3.3 Linking number on an arbitrary immersion of the torus

We now consider an arbitrary immersion of the torus in 3d. Its self-intersection curve $\Delta$ is characterized by its winding numbers. To define them with sign, it is necessary to give an orientation to $\Delta$. For a point $x$ in $\Delta$, we define two vectors $\xi_1$ and $\xi_2$, $\xi_i$ being normal to the sheet containing the two preimages $\Delta_i$ $(i = 1, 2)$. Unless the two sheets are parallel (which does not happen generically), the orientation $e$ is unambiguously defined by $\xi_1 \times \xi_2$. Note that we can consider that this is the orientation of $\Delta_1$ (while that of $\Delta_2$ will be opposite).

With this orientation, if the self-intersection curve $\Delta$ is connected, its winding numbers are noted $\boldsymbol{\delta} = (\delta_x, \delta_y)$ where $\delta_x$ and $\delta_y$ are positive or negative integers. Suppose that the two components of

$$\boldsymbol{\delta} = (nq, mq), \tag{35}$$

are multiples of $q$, the greatest common divisor, $q = \gcd(\delta_x, \delta_y) > 0$. In this case, we shall say that the immersion is "$q$-perfect". For example, with this definition, the double covered torus (previous paragraph) is "1-perfect" since its connected self-intersection curve has winding numbers $(1, 0)$, $q = 1$.

More generally, if the self-intersection curve $\Delta$ is *not* connected and has $m$ components, the integer $q$ is defined as the greatest common divisor of the set of all winding numbers,

$$(\delta_x^1, \delta_y^1, \ldots, \delta_x^m, \delta_y^m). \tag{36}$$

This is the most general definition of a "$q$-perfect" immersion. If the set is empty or if all the winding numbers are zero, we say that the immersion is "$\infty$-perfect". Note that an immersion that is 6-perfect, for example, is also 2 and 3-perfect but the converse, is of course, not true.

As a consequence of this definition, a loop $l_a$ with winding numbers $\boldsymbol{a}$ will have $\boldsymbol{a} \times \boldsymbol{\delta}$ signed crossings with the self-intersection curve, which will be a multiple of $q$. The loop $l_a$ crosses $\Delta_1$ in a certain number of points, $x_1, \ldots, x_s$, each equipped with a sign (because $l_a$ and $\Delta_1$ are orientated). Similarly, a second loop $l_b$ crosses $\Delta_2$ in $y_1, \ldots, y_r$. These points correspond to some points on $\Delta_1$, which are sent to the same points on $\Delta$ by the immersion. If $l_a$ passes through one of these points on $\Delta_1$, there will be an intersection of $l_a$ and $l_b$ on $\Delta$. We can first consider that $l_a$ avoids these points so that the $x$'s and $y'$s are placed in some order, going along $\Delta_1$. One can define some particular order, *e.g.* all the $x$'s to the left of all the $y$'s (or it can be empty), and call it the "normal order", for which we have (by definition) a "disorder" number,

$$\#(l_a, l_b) = 0. \tag{37}$$

Now for a generic configuration, the disorder number $\#(l_a, l_b)$ is defined as the signed number of permutations of the $x$'s and $y$'s. Because of the torus geometry, one can move $l_a$ to the right of the set $y_1, \ldots, y_r$ to realize a different order. This corresponds to a certain number of permutations of the normal order, that is a multiple of $q$,

$$\#(l_a, l_b) = nq, n \in \mathbb{Z}. \tag{38}$$

However, it is the same state as the original one, so to match the disorder numbers, we have to define it modulo $q$,

$$\#(l_a, l_b) = 0 \pmod{q}. \tag{39}$$

Now each crossing of $l_a$ with a point $y_j$ permutes the order by a transposition, $x_i y_j \to y_j x_i$, it corresponds to a crossing of the two lines $l_a$ and $l_b$ on the immersion and a jump of the linking number, which we correct by counting the permutation, $\#(l_a, l_b) = \pm 1 \pmod{q}$.

Consider the linking number,

$$\mathrm{LK}(\boldsymbol{a}, \boldsymbol{b}) = \mathrm{lk}(L_a, L_b) + \sharp(l_a, l_b) \pmod{q}, \tag{40}$$

which is a half-integer number: $\boldsymbol{a}$ labels the homotopy class of $l_a$ (the closed loop on the surface), whereas $L_a$ is its image on the immersion. It is now a well-defined integer modulo $q$ for a $q$-perfect immersion, that depends only on the homotopy classes of the loops, not the loops themselves. If the immersion is $\infty$-perfect, the linking number is well-defined for any $q$, in particular for an embedding for which $\sharp(l_a, l_b) = 0$ and the expression reduces to the standard linking number.

## 3.4 A topological invariant

We now explain that the integer linking number given by Eq. (40) is a topological invariant of the immersed surface. We call two immersions $\varphi_1$ and $\varphi_2$ "$q$-perfect regular homotopic" if

there exists a regular[3] homotopy $\varphi_t$ with $t \in [0, 1]$, which coincides with the two immersions at $t = 0$ and $t = 1$, with a finite number of reconnection points of the self-intersection curves, and such that an arbitrary immersion $\varphi_t$ is $q$-perfect. In other words, the deformation of the initial immersion should be such that the self-intersection curves, although they may reconnect in a finite number of points (the number of components $m$ may not be conserved), remain $q$-perfect at all "times". In this case, the half-integer linking number (40) is a topological invariant. The existence of different immersions for which it is possible to compute the topological invariant given by Eq. 40 will tell us, if these numbers differ, that there is no $q$-perfect regular homotopy between them, and thus gives us some information on the evolution of the winding properties of the self-intersection curve upon regular homotopy. The classification of immersions up to $q$-perfect regular homotopies is an interesting problem. We now consider the example of the torus.

## 4   Two classes of immersed torus

We now consider immersions of the torus in 3d, up to regular homotopies and reparametrizations: it is known [25] that there are two classes of immersed torus (contrary to the single regular homotopy class of the 2-spheres), a representative of each is shown in Fig. 6 and will be described below. The regular homotopies conserve an invariant, which is called the Arf-Kervaire invariant of the immersion $s$,

$$\mathrm{Arf}(s) = q(x)q(y) \pmod 2, \tag{41}$$

where $x$ and $y$ are two independent cycles on the surface and the function $q(x)$ is defined as follows. $q(x)$ measures the integer number of $2\pi$-turns of a "thickened" cycle $x$ (also called the "tubular neighbourhood" or $x$). A "thickened" cycle is an infinitesimal "ribbon" constructed on the surface with $x$ as one edge and a parallel copy of $x$ as the second edge. The "ribbon" has a well-defined frame, a collection of three orthogonal unit vectors that measures its local orientation (tangent to the cycle, normal to it and belonging to the surface, and binormal, *i.e.* perpendicular to the surface). The frame defines at each point of the ribbon an $SO(3)$ matrix which is obtained by rotating a reference frame onto the local frame thus defined. In turn, this defines a continuous application from the cycle $S^1$ onto $SO(3)$. We know that there are two homotopy classes of such maps corresponding to $\pi_1[SO(3)] = \mathbb{Z}_2$: the generator corresponds to the Frenet-Serret frame of the simple circle. When thickened, it gives an untwisted ribbon with $q(x) = 0$. The trivial element on the other hand is the circle run over twice, or, as a ribbon, the twisted ribbon (by a $2\pi$-turn); *i.e.* $q(x) = 1$.

An alternative viewpoint is to consider the two edges of the ribbon, one is the original cycle $x$ and the other one is its "parallel" copy, $x'$. It is easy to see that when the ribbon is twisted by a $2\pi$-turn, its two edges $x$ and $x'$ are linked with linking coefficient $\mathrm{lk}(x, x') = \pm 1$ (with a sign depending on the orientation). Whereas when the ribbon makes no turn, the two cycles are obviously unlinked and $\mathrm{lk}(x, x') = 0$. So that we can also define

$$q(x) = \mathrm{lk}(x, x'), \tag{42}$$

and $q(x)$ is either 0 or 1 by definition. We have therefore two types of ribbons that we cannot continuously deform into each other. They allow to define two inequivalent immersed torus, characterized by the Arf-Kervaire invariant, Eq. (41), that cannot be connected by regular homotopy [25].

---

[3]A *regular* homotopy is a homotopy between immersions, *i.e.* the immersion must have tangent vectors well-defined at every point and all "times" of the deformation.

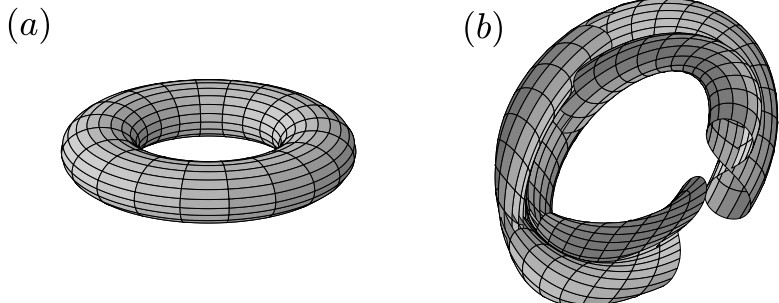

Figure 6: Two inequivalent classes of immersed torus up to regular homotopies: (a) standard, embedded, Arf-Kervaire 0, torus; (b) immersion, Arf-Kervaire 1, self-linked torus, which is similar to that of Fig. 5 with a twist.

The first torus [Fig. 6 (a)] is the *standard* embedded torus in 3d. It has two cycles $x$ and $y$ which satisfy $q(x) = q(y) = 0$, which means that the two corresponding ribbons are untwisted (or their edges unlinked): it has an Arf-Kervaire invariant of zero.

The second torus [Fig. 6 (b)] has two cycles $x$ and $y$ which satisfy $q(x) = q(y) = 1$, so that the two ribbons are twisted, and we say that the cycles are "self-linked". It has an Arf-Kervaire invariant of one. The simplest example of such a torus consists of rotating a lemniscate (a planar curve with the shape of the number eight) by $2\pi$ while moving its double point along a closed circle. This generates a torus which is immersed in space, with a self-intersection line which originates in the double point of the lemniscate. The first cycle $x$ consists of following a single lemniscate, avoiding its double point. The corresponding thickened cycle is twisted by $2\pi$. The second cycle $y$ consists of following a point at the top of the lemniscate while rotating it, it is equivalent to a $(1,1)$ cycle on a regular torus and hence is "self-linked". This is therefore an example of an immersed torus with Arf-Kervaire invariant one. The self-intersection curve has winding numbers $(1,0)$, in this instance (the immersion is 1-perfect). In the following, we will need to construct a variant of this surface with a self-intersection curve that has no such winding components.

It is also known that the Arf-Kervaire invariant is not only a homotopy invariant but also a regular cobordism invariant, and hence an invariant of higher homotopy groups[4]. A consequence is that the present immersion represents the nontrivial element of the stable homotopy group $\Pi_2 = \pi_{2+n}(S^n) = \mathbb{Z}_2$ when $n > 3$. We will see that one color exchange invariant is an invariant distinguishing two chiral subclasses of the nontrivial class of $\Pi_2$.

## 5 Invariant as a linking number

We now construct an immersed torus, the Konstantinov torus, that is *self-linked* (Arf-Kervaire invariant one) and has a self-intersection curve with no winding components and is, hence, "$\infty$-perfect". On this immersion, the invariant $I_2$ appears as a linking number.

For a $q$-perfect immersion, we have seen in section 3.3 that the linking numbers (with disorder parameters) are well-defined modulo $q$. Expand over the basis cycles to obtain,

$$\text{LK}(\boldsymbol{a}, \boldsymbol{b}) = a_x b_x \text{lk}(\boldsymbol{x}, \boldsymbol{x}) + a_y b_y \text{lk}(\boldsymbol{y}, \boldsymbol{y}) + a_x b_y \text{lk}(\boldsymbol{x}, \boldsymbol{y}) + a_y b_x \text{lk}(\boldsymbol{y}, \boldsymbol{x}) \pmod{q}. \quad (43)$$

For a self-linked immersion $q(\boldsymbol{x}) = \text{lk}(\boldsymbol{x}, \boldsymbol{x}) = 1$, $q(\boldsymbol{y}) = \text{lk}(\boldsymbol{y}, \boldsymbol{y}) = 1$ and $\text{lk}(\boldsymbol{x}, \boldsymbol{y}) = 1/2$.

---

[4]See, for example, B. A. Dubrovin, A. T. Fomenko and S. P. Novikov, Modern geometry- methods and applications, Vol. 2, page 216. Springer, Graduate Texts in Mathematics.

Multiply by 2 to obtain

$$2\text{LK}(\boldsymbol{a}, \boldsymbol{b}) = 2a_x b_x + 2a_y b_y + a_x b_y + a_y b_x \pmod{2q}, \tag{44}$$

which is precisely the Eq. (14) for $I_2$, provided that $q = 2$. The immersion of Fig. 6 (b) is 1-perfect ($q = 1$), so that $2\text{LK}(\boldsymbol{a}, \boldsymbol{b})$ could be 0 or 1 and will be always 1 for the two sectors $I_2 = \pm 1$. To distinguish them, it is thus necessary to consider a different immersion with $q \geq 2$.

## 5.1 Construction of a Arf-Kervaire one, "∞-perfect", immersed torus

The Konstantinov torus is an immersed torus with an Arf-Kervaire invariant of one which is ∞-perfect, originally considered by N. N. Konstantinov and described in details in Ref. [26]. For completeness, we explain how it is constructed. Consider the two sheets $k$ and $k'$ in Fig. 7: each is a (curvilinear) hexagon with six edges. $k$ and $k'$ are glued together along the three segments AB, CD, EF. After gluing, there remains three free edges from $k$ and three free edges from $k'$ which form a single closed curve. The two sheets overlap and are now deformed in the $z$ direction, perpendicular to the plane of the figure. In the deformation, we push the surface towards $z \geq 0$, keeping its single edge precisely in the original plane $z = 0$. In this way, we remove most of its overlap, except for the three arcs $(af)$, $(bc)$ and $(de)$, which are therefore part of the self-intersection curve. It is clear from the figure that the sheet $k$ which is above $k'$ to the left of $(af)$ goes under $k'$ in the central region of the figure. We immediately see that the resulting surface has a $2\pi/3$ symmetry and is chiral (no mirror plane through AB for instance), the two chiralities being obtained by selecting which of the two sheets $k$ or $k'$ is above the other in the central region, giving either Fig. 7 (a) or Fig. 7 (b).

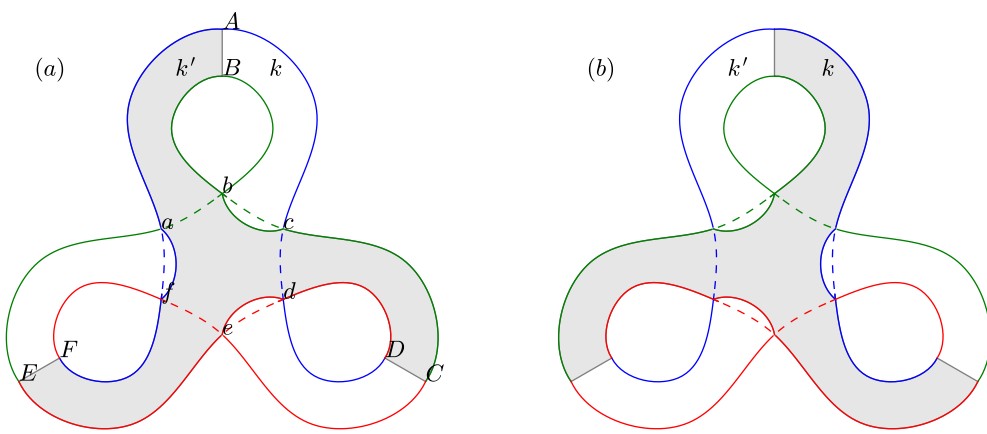

Figure 7: (a) Part of the Konstantinov torus (it is an immersed torus in 3d, here with a disk removed along the free edge): two "hexagonal" sheets $k$ ($ADCFEB$) and $k'$ ($ABCDEF$) glued together along $AB$, $CD$ and $EF$, giving a sheet $k \cup k'$ with a single edge at $z = 0$ (the rest of the surface being at $z > 0$). There are three intersections along the arcs $(af)$, $(bc)$, and $(ed)$ (solid lines), where $k$ crosses under $k'$. (b) The chiral image of (a): $k'$ crosses under $k$. The impossibility to connect these two immersions (a) and (b) by a 2-perfect regular homotopy provides a topological interpretation of the color exchange invariant $I_2$ (see section 5.2).

In order to close the surface, we need to attach a disk to the free edge. To explain how we do this, it is convenient to attach the disk through a cylinder, which should be seen in

the $z \leq 0$ part of 3d space. Fig. 8 gives several cuts of the cylinder at constant $z \leq 0$, from $z = 0$ (subfigure (9)) attached to the free edge of Fig. 7 (a), to a finite $z < 0$ (subfigure (1)) attached to a simple disk. It shows how the cylinder is folded and where the self-intersections are. To visualize the self-intersection curve, we can start from the point $a$ at $z = 0$ and follow where the intersection point goes on the cylinder in the sequence (1)-(9) of Fig. 8. With the present choice, $a$ goes over to $b$. By $2\pi/3$ symmetry, $c$ goes to $d$, and $e$ to $f$ so that the self-intersection curve is *connected* through the remaining arcs $(bc)$, $(de)$ and $(fa)$ (there is no need to discuss the additional components on the cylinder which will not play a role in the winding properties). We can also see that there are four triple points.

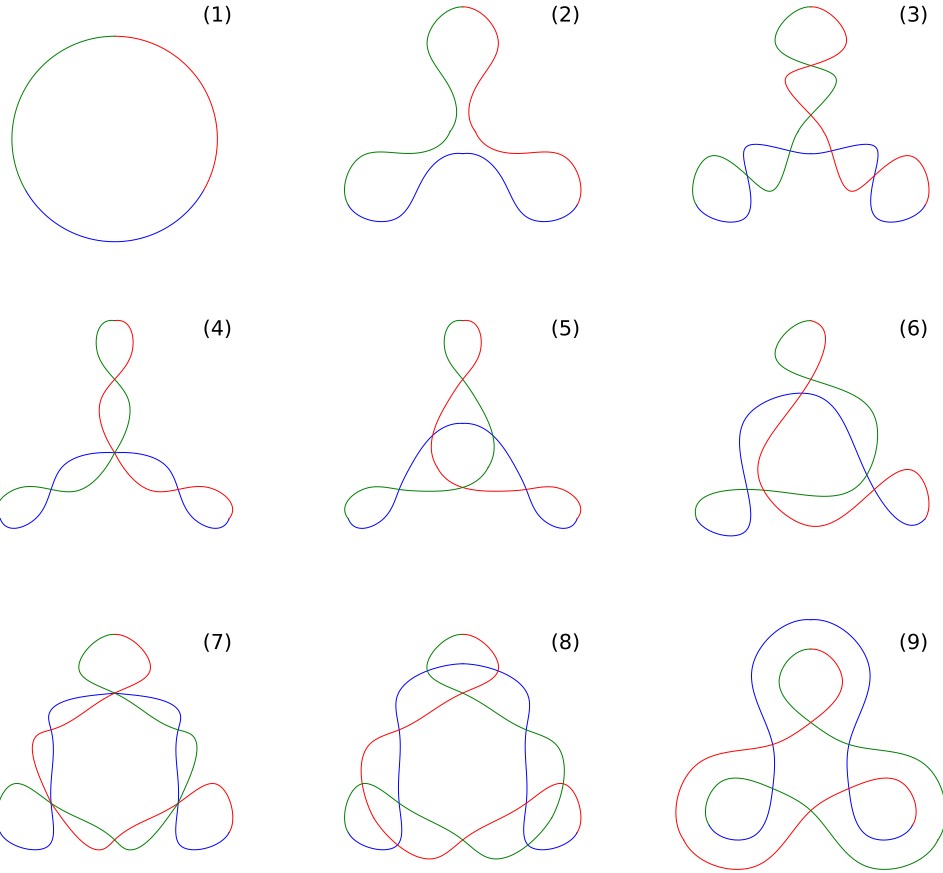

Figure 8: Several cuts of the cylinder, at different $z$, whose first free edge (1) will be connected to the disk and the second free edge (9) to the surface $k \cup k'$ along its $z = 0$ edge (Fig. 7). Each crossing is a double point and there are 4 triple points in (4) and (7). In between steps (5) and (7), the mirror plane perpendicular to the plane of the figure is broken: a chirality is chosen which must map the chirality of the sheet $k \cup k'$.

The surface thus constructed is a torus. It is evident from the fact that the three sheets, $k$, $k'$ and the disk can be viewed as three hexagons, on which a hexagonal lattice can be drawn: construct a single hexagon on each (with six dangling bonds out of it) and connect $k$ and $k'$ using three of these dangling bonds. The six remaining bonds of $k$ and $k'$ will be used to connect the six free bonds of the disk. In this case, we have $3 \times 6 = 18$ vertices, 6 edges on each piece and 6 edges for the three pairs of pieces, $3 \times 6 + 3 \times 3 = 27$ edges and 9 faces. The

Euler characteristic is $\chi = V - E + F = 0$, that of a torus, indeed.

The basis cycles (the basis elements of $H_1(T^2, \mathbb{Z})$) of such an immersion are self-linked. There are three equivalent cycles (related by $2\pi/3$ symmetry) which we call $x$, $y$ and $z$. Each has the shape of a figure eight and can be all contained in $k \cup k'$: the cycle $x$ starts from the middle of the segment AB, for instance, and goes over to CD on $k'$ and from CD to AB on $k$. Note that indeed if we cut the surface along $x$, it does not separate the surface into two pieces, it is therefore not the border of a surface and is a nontrivial cycle. On a torus, there are two primitive cycles and they intersect once. The second one is therefore $y$ (which goes from CD to EF on $k$ and back on $k'$), or equivalently $z$. Consider the tubular neighbourhood of $x$. The ribbon has the shape of a lemniscate (with its double point avoided in the third dimension). It is easy to see by cutting the ribbon along its middle line that the two ribbons thus obtained are *linked*, $\mathrm{lk}(x, x') = 1$, we say that the cycle is self-linked. If, however, we look at the frame of the ribbon, we see that the binormal vector is always close to the $z$-axis whereas the normal and tangent vectors do not do a full turn. This element is the trivial element of $\pi_1[SO(3)]$: it can be folded as two circles one above the other and corresponds to a $4\pi$ rotation (trivial element). By symmetry, the same argument applies to $y$ or $z$, so that $q(x) = q(y) = q(z) = 1$. The immersion has therefore an Arf-Kervaire invariant of one and cannot be transformed by regular homotopy onto the standard torus [25]. The self-intersection curve on the torus with the hole does not wind around, it is obvious that each arc (*e.g.* $(af)$) does not have an overlap with $x$, $y$ or $z$: the self-intersection curve is homotopically trivial (all other components are closed on the disk and are similarly trivial) and the immersion is therefore $\infty$-perfect. We have thus explained the construction and the properties of a Arf-Kervaire invariant one, $\infty$-perfect, torus.

The Konstantinov torus is, in particular, 2-perfect, and Eq. (44) holds modulo 4. The invariant $I_2 = \pm 1$ is thus visualized as the obstruction to unlink the cycles on the Konstantinov torus.

## 5.2 Interpretation as a topological obstruction

We can provide a topological interpretation of the color exchange invariant $I_2$, following section 3.4.

Consider the two Konstantinov torus that are image of each other by a mirror plane given in Fig. 7 ($a$) and Fig. 7 ($b$). Note that they both have an Arf-Kervaire invariant of one, so that there exists a regular homotopy between them.

We have just explained that it is possible to construct configurations with $2LK = +1$ (mod 4) ($I_2 = +1$ in section 2.1) on a given Konstantinov torus and that any 2-perfect regular homotopy will not change this number. Since LK is a chiral number, it is changed in its opposite by reflections and the chiral image of the immersion will have $2LK = -1$. As we have seen in section 3.4, 2LK is conserved modulo 4 by a 2-perfect regular homotopy. As a consequence, there is no 2-perfect regular homotopy that will connect the two chiral Konstantinov immersions: they form new chiral classes by themselves for 2-perfect homotopies, characterized by $2LK = \pm 1$.

The nontrivial class of $\Pi_2$ thus admits subclasses, *i.e.* immersions that cannot be transformed into one another by 2-perfect regular homotopies. The color invariant $I_2 = \pm 1$ thus reflects this obstruction between two immersions, up to such deformations.

# 6 Conclusion

We have explained that the invariants $I_2$, $I_2'$ and $I_2''$ associated with color exchange can be visualized as a linking of the Kempe loops, when properly immersed in 3d: the obstruction to the dynamics I, described by $I_2 = \pm 1$, needs to be visualized as linked winding loops on a self-linked, Arf-Kervaire one, immersed torus, while that of dynamics II described by $I_2' = 0$ or 1, or $I_2'' = \pm 1$, appears as linking on the standard embedded torus.

An equivalence has emerged between invariants of dynamics and finer classes of homotopies or cobordism, which are the equivalence classes of "$q$-perfect" regular homotopies. It reflects the possibility (or not) to connect immersions by regular homotopy or cobordism *without modifying the winding properties of the self-intersection curves*. This is this mathematical property which is equivalent (in the sense of being described by the same invariants) to the obstructions in the physical dynamics of colors.

In this sense, the obstruction in color dynamics I is thus explained by the topological obstruction to transform the immersion given in Fig. 7 ($a$) into its chiral image (Fig. 7 ($b$)), up to the continuous deformation we have defined. The color invariant thus arises as a new obstruction in $\Pi_2$, resulting in finer chiral subclasses.

While we have explained the correspondence between $I_2$, $I_2'$ and linking numbers on immersions (and $I_1$ as the parity of the number of intersections of winding loops), it is possible that the invariants $I_3^{\pm}$, $I_3^{\pm'}$ or, more generally, invariants of other similarly constrained problems, could be expressed in terms of higher invariants which are not functions of pairwise linking numbers but more general invariants or intersection forms.

# 7 Perspectives and open problems

The existence of disconnected classes is a property that depends on the topology of the surface. While the torus has many classes, the projective plane, for example, has only one [11]. Generalizations to other surfaces $M$ may be possible for either *nonorientable* surfaces, or orientable surfaces with genus $g > 1$ (for which $2g$ noncontractible cycles exist). It would involve replacing the two winding numbers of a given loop by an element of the (in general, nonabelian) group $\pi_1(M)$. Following the present work, a possible outcome is a finer classification of immersions of the surfaces. While these problems are not directly motivated by applications, we nevertheless note the relevance of three-colored states in large magnetic molecules having the topology of the sphere ($g = 0$) [8].

- On the *nonorientable* Klein bottle, the dynamics of 3-colored states on a hexagonal lattice also has Kempe sectors [20]. Immersions of the Klein bottle (or, more generally, of a nonorientable surface), are characterized by the Arf-Brown invariant, defined modulo 8 (reflecting the possibilities of twists by $\pi$-turn), instead of modulo 2, as in Eq. (41). This provides further invariant properties of the immersions, representing higher-homotopy group elements.

- Other examples of color problems can be constructed on surfaces $M$ with higher genus $g > 1$. While it is possible to tile the Euclidean plane with regular hexagons (with angles $2\pi/3$), the hyperbolic plane can be tiled with curvilinear hexagons with angles $2\pi/n$, where $n > 3$ is an integer (giving periodic hyperbolic lattices with Schläfli symbol $\{6, n\}$). Since, at each vertex, $n$ edges meet, the lattice can be colored with a minimal number of $n$ colors. The problem is that of the equivalence classes of $n$-colorings on compact surfaces constructed from $\{6, n\}$ hyperbolic lattices. Another sequence is given by lattices with Schläfli symbols $\{p, 3\}$ (the end point of the sequence, $\{\infty, 3\}$, being

a simple loop-free Bethe lattice with coordination number 3). In this case, three edges meet at each vertex, so the lattice can be 3-colored, and corresponds to hyperbolic tilings with heptagons, octogons etc., depending on $p$. All these 3-colored hyperbolic lattices may have new invariants depending on their genus. Such special graphs have not been studied, to our knowledge, but would shed light on the effect of the topology of the surface.

# Acknowledgements

P. M. A. would like to acknowledge support from grant RNF 21-11-00010.

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
