# Peer review of "Topological interpretation of color exchange invariants: hexagonal lattice on a torus"

_SciPost Physics, doi:SciPost Phys. 10, 042 (2021)_

## Round 1 · Referee Report · Anonymous (Referee 1) · 2020-10-21

Report

This paper studies the connectivity of configuration space of the three coloring
model on the honeycomb lattice under two types of dynamics. This is
quite an interesting topic. For practitioners of all sorts of
numerical investigations, the existence of disjointed sectors is an
important feature when interpreting the reliability of the data thus
obtained. The organisation of a locally constrained configuration space is of course also an interesting topic in its own right. While my knowledge of the mathematical side of things is quite
limited, the authors seem to have quite a deep visibility into the
state of the art in discrete mathematics.

The manuscript contains a detailed account of the invariants
identified, backed up with connections to and concepts from the mathematics
literature. These are combined with the results from exhaustive
searches of configuration space for finite-size systems.

The results look reasonable, and I have not seen anything that looks
wrong. However, I should state that I am not an expert on the
mathematical side of things, so that I cannot judge the validity from
a formal perspective.

I was a little bit disappointed by a couple of aspects of the
manuscript. Regarding the introduction, I find the amount of
background given is somewhat limited. Given the authors are among the
world experts in this specific field, I feel the reader could benefit
from a more thorough exposition of the field. For instance, there is a
long-established literature on dimer models, referred to in passing as
`two-color (dimer) models' (page 2). I think this is not really
correct, as for a lattice of coordination z, there are z-1 identical
colors meeting at a vertex in a dimer model, so this is really not a
coloring problem. Also, the level of referencing seems quite low, so I
was wondering if there were really practically no precursors to this
work?

To some degree related is a somewhat weak motivation of the two types
of dynamics which are being considered. Are these two (labelled
`dynamics I' and `dynamics II') the only ones? The only natural ones?
The only tractable ones? Perhaps I can suggest to the
authors to go over the text again with the eyes of an uninitiated
reader and to construct a higher-level narrative.

Finally, the title might raise expectations a little bit beyond the
actual content of the manuscript. At least when I read it, I was
expecting something more general about coloring models than the study
of specifically the three-coloring honeycomb model. Of course, the
ideas developed here may well be of use more broadly, but I do not
think that this is more than a general perspective at this point.

I find the manuscript to be publishable overall, as the progress
reported here, on a difficult problem, is definitely nonzero. If the
authors can improve on the aspects above, I think publication in
Scipost Core would be appropriate.
  • validity: -
  • significance: -
  • originality: -
  • clarity: -
  • formatting: -
  • grammar: -

Author:  Olivier Cépas  on 2021-01-22  [id 1173]

(in reply to Report 1 on 2020-10-21)

We would like to thank the first referee for her/his careful reading, and suggestions of change.

"Regarding the introduction, I find the amount of background given is somewhat limited."

We have corrected this point, and added some references on similar topics (dimer models), or broader mathematical perspectives on dynamics, related to the issue of dynamics II (see below).

"For instance, there is a
long-established literature on dimer models, referred to in passing as
`two-color (dimer) models' (page 2). I think this is not really
correct, as for a lattice of coordination z, there are z-1 identical
colors meeting at a vertex in a dimer model, so this is really not a
coloring problem. Also, the level of referencing seems quite low, so I
was wondering if there were really practically no precursors to this
work?"

The referee is right that a dimer model is not really a "two-color" model as we erroneously stated. Precursors of the work have been cited (see e.g. the papers by Mohar and Salas, Sokal, and Huse and Rutenberg) but we have added some more similar works, e.g. on dimer models.

"To some degree related is a somewhat weak motivation of the two types
of dynamics which are being considered. Are these two (labelled
dynamicsI'anddynamics II') the only ones? The only natural ones?
The only tractable ones? "

We have also corrected this point. The dynamics I and II are not the only ones, not the only tractable ones: any further constraint can in fact be considered and we hope this point is now clear in the manuscript. The only natural one is dynamics I. It is not clear that dynamics II could have some physical origin, but it is an interesting example since it generates some new nontrivial sectors, the invariants of which we could find.

The title has been changed.

---

## Round 1 · Referee Report · Anonymous (Referee 2) · 2020-11-9

Strengths

1) The topic is rather difficult, and falls amid Discrete Mathematics, Algebraic Topology, and Statistical Mechanics. 2) It has strong relevance in Monte Carlo studies of models in Statistical Mechanics.

Weaknesses

1) The title seems to claim a stronger result that the actual results obtained by the authors. They should reconsider finding a more precise title. 2) Some changes are needed to avoid misunderstandings with readers with a more mathematical background.

Report

The authors study the number of Kempe equivalence classes for the edge 3-coloring of the hexagonal lattice. They consider two distinct dynamics for this colouring problem, and find some invariants for each dynamic. For dynamic II, some invariant sectors have been found empirically via numerical simulations. All these invariants for both dynamics are shown to be related to certain topological invariants. This manuscript falls close to several areas of research: i.e., Discrete Mathematics, Algebraic Topology, and Statistical Mechanics.
Unfortunately, my knowledge of Algebraic Topology is limited, so I refrain from making technical comments on sections 3, 4 and 5.

The posed problem is rather difficult and very important in Monte Carlo simulations. Given a dynamics related to Kempe
changes, and such that there exist at least two Kempe equivalence classes, then the corresponding Markov chain fails to be irreducible (ergodic), and the convergence of the probability distribution to the stationary one, as time grows to infinity, is no longer guaranteed.

The paper is clear enough, well organized and written, but there are some points that in my opinion might be improved and/or revised (see list below).

In summary, I think this manuscript is a relevant addition to this notoriously difficult problem. The search for new invariants in dynamics related to Kempe chains opens a path that could help others to solve open vertex and/or edge colouring problems. After the authors reconsider the points raised above, I consider this manuscript satisfies the acceptance criteria of SciPost Physics.

Requested changes

1) The title should be replaced by a more precise statement about the results obtained in the paper. The original title claims a much stronger result that the results obtained by the authors. 2) At the end of the first paragraph on page 2, they say "and loop dynamics or gauge theories in physics,"; but there are other well-know relations between graph coloring problems and physical systems: e.g., the zero-temperature q-state Potts antiferromagnets on certain graphs (in this case, we have a vertex proper q-colouring of that graph). 3) In the second paragraph on page 2, they study "The colouring of the edges of the regular hexagonal lattice with three colours", but there are actually several definitions of a (proper) edge q-colouring: e.g., a) At each vertex of the graph, the colours of the incident edges to that vertex should be distinct. (This is the most common edge-colouring problem). b) On each face of the graph (embedded on some surface), the edges bounding that face should have distinct colours. This is the problem considered in Ref. [12] where the graph is any 6-regular triangulation of the torus. Even though the authors study 3-color edge-colourings of a hexagonal lattice on the torus, so it is more or less clear that they are referring to definition a), they should explain better to which edge-colouring problem they are dealing with. 4) On top of page 4, they consider two distinct dynamics, but it is not clear why these are the most relevant ones, or if there are any more interesting dynamics. 5) In Section 5.1 (as well as in the abstract), they consider "the 2d hexagonal lattice", but in graph theory the physical space-time dimension is not a well-defined concept. One has to consider either planar or non-planar graphs. The former ones can be embedded on a surface, and the latter ones can only be embedded on surfaces of higher genus. It is clear that the authors consider subsets of the regular hexagonal lattice wrapped on a torus. From a physical point of view this class of graphs are of most importance. 6) I would suggest to add two previous Fisk's papers: Fisk S, 1973, Adv. Math. 11, 326, and Fisk S, 1977, Adv. Math. 24, 298, as ref. [14] is based on both of them. 7) Is it possible to extend the techniques or the results obtained in this manuscript to other vertex colourings with an unknown number of Kempe classes? 8) Page 2. First sentence after the section title. It reads "in its attempt" and it should read "in his attempt". 9) On page 9 and the following ones, they use $lk$ in math mode, and I consider it would be clearer to use $\mathrm{lk}$. 10) On page 12, Eq. (39) and the following lines. The same happens to $LK$, that is probably less clear than $\mathrm{LK}$. 11) On page 12, Eq. (40) and the following lines. The same happens to $Ar f(s)$, that is probably less clear than $\mathrm{Arf}(s)$.

  • validity: high
  • significance: high
  • originality: top
  • clarity: high
  • formatting: excellent
  • grammar: perfect

Author:  Olivier Cépas  on 2021-01-22  [id 1174]

(in reply to Report 2 on 2020-11-09)

We would to like to thank the referee for her/his careful report, and suggestions of improvement. Here is a detailed response for the requested changes.

1) This is true (a point also raised by referee 1), and we have specified more precisely what the problem is in the title. 2) The end of the first paragraph has been changed along the referee's remark. 3) The model is now properly defined in the introduction. 4) This remark has been raised also by the other referees, and a paragraph has been added to explain the choice, content and background of the two dynamics, as well as two broad references. 5) We have changed the presentation in second paragraph of the introduction, to make it clear that a torus topology was considered (nonplanar graphs). 6) The two previous Fisk's papers have been added. 7) It is certainly possible to study other vertex colorings along these lines and would be interesting to understand what is specific to each model. It seems that linking numbers are quite generic. 8) its attempt -> his attempt corrected. 9), 10), 11) lk, LK, and Arf in math mode have been changed.

---

## Round 1 · Referee Report · Anonymous (Referee 3) · 2021-1-11

Report

Many models in frustrated magnetism can be mapped to coloring problems on lattices. Dynamics based on loop-like updates of coloring configurations are important for Monte Carlo simulation of these problems and in mathematical treatments. However, in many cases such updates are non-ergodic, and classifying the different dynamical sectors may be highly nontrivial.

In a previous paper, these authors defined a set of invariants for the 3-coloring model on the honeycomb lattice, with periodic boundary conditions, which they argued to be complete in a certain sense. In the present paper, they give a topological interpretation of some these invariants, in terms of linking numbers of Kempe chains (chains of alternating color) when the torus is embedded or immersed in three dimensions.

This result seems (to a nonexpert) both interesting and nontrivial and I am happy to recommend publication of the paper. Below are some suggestions and queries that the authors may wish to consider.

Although the paper is well written, there is little explanation of background, even of definitions, which means it is not very accessible for a non-specialist. A partial remedy would be to tell the reader more clearly to look in the authors’ earlier paper [15] for explanations of the dynamics, winding numbers and invariants. However, the authors refer early on to “color exchange” and “the dynamics” and in order to be self-contained it would be helpful to explain what is meant. For example, there are physical contexts where the relevant dynamics is restricted to color exchanges on short loops (e.g. hexagons). It would be worth making explicit that here the important exchanges are instead along winding loops. It might also be useful to repeat the basic figures introducing the model from [15].

The authors introduce a modified “dynamics II”. The definition is algebraic. Can they state the physical interpretation? (E.g. we are only allowed to flip loops with even winding in the x direction.) What is the motivation for this definition?

The authors distinguish between stable invariants, which give a fixed finite number of sectors, and additional non-ergodic sectors whose number depends on L that are due instead to steric constraints at fixed size L. In order to clarify the importance of the stable invariants: can it be stated that the additional sectors (due to steric constraints) contain only a vanishing fraction of the entropy of the honeycomb lattice coloring model in the thermodynamic limit?

The relation to linking on the torus is made at the level of the expressions for the invariants. Is it possible to give a more intuitive “pictorial” explanation, at the level of the color exchange update, for why the linking number (as defined here, to include 1/2 integer values) cannot change? If not, perhaps the authors could give the reader some of the intuition that led them to this identification.
  • validity: -
  • significance: -
  • originality: -
  • clarity: -
  • formatting: -
  • grammar: -

Author:  Olivier Cépas  on 2021-01-22  [id 1175]

(in reply to Report 3 on 2021-01-11)

We would like to thank the third referee for her/his careful report and clear suggestions.

"Although the paper is well written, there is little explanation of background, even of definitions, which means it is not very accessible for a non-specialist. A partial remedy would be to tell the reader more clearly to look in the authors’ earlier paper [15] for explanations of the dynamics, winding numbers and invariants. However, the authors refer early on to “color exchange” and “the dynamics” and in order to be self-contained it would be helpful to explain what is meant. For example, there are physical contexts where the relevant dynamics is restricted to color exchanges on short loops (e.g. hexagons). It would be worth making explicit that here the important exchanges are instead along winding loops. It might also be useful to repeat the basic figures introducing the model from [15]."

It is true that the previous version of the manuscript relied on the definitions published in the earlier paper. To make it more self-contained, we have added the basic definitions and two figures. We hope it is now more directly readable. "Color exchange" is now broadly defined in the second sentence of the paper. The distinction between short and long loops is made clear in the introduction (end of the third paragraph). More background is given (examples of non-ergodic dynamics in short-loop dimer models).

"The authors introduce a modified “dynamics II”. The definition is algebraic. Can they state the physical interpretation? (E.g. we are only allowed to flip loops with even winding in the x direction.) What is the motivation for this definition?"

Dynamics II is not motivated by the physical color problem but is an interesting example that some further constraints may lead to additional nontrivial sectors. Furthermore, the resulting invariants are very simple, and give a partial answer to the "more intuitive pictorial explanation" (last remark of the referee). To put it in a broader context, we have added the references to papers by Arnold and Vassiliev and a short sentence in the introduction.

"The authors distinguish between stable invariants, which give a fixed finite number of sectors, and additional non-ergodic sectors whose number depends on L that are due instead to steric constraints at fixed size L. In order to clarify the importance of the stable invariants: can it be stated that the additional sectors (due to steric constraints) contain only a vanishing fraction of the entropy of the honeycomb lattice coloring model in the thermodynamic limit?"

If we take randomly a state with given winding numbers, the probability for this state to belong to an unstable sector is vanishingly small in the dilute limit. This is explained in the previous paper.
Usually (in similar dimer models), the probability for a color state to belong to a high winding sector is itself exponentially small, $\exp(-n^2)$ (here $n$ would be as defined in the caption of table 1). The entropy thus decreases as $S_0-\alpha n^2$, but is probably non-vanishing. Quite surprisingly, the additional sectors due to steric constraints may have quite small $n$ (see Fig. 12 of our previous paper). The question is therefore not simple, but could be further investigated.

"The relation to linking on the torus is made at the level of the expressions for the invariants. Is it possible to give a more intuitive “pictorial” explanation, at the level of the color exchange update, for why the linking number (as defined here, to include 1/2 integer values) cannot change? If not, perhaps the authors could give the reader some of the intuition that led them to this identification."

It is not easy to visualize the processes themselves since the loops can be cut and reconnected but probably not impossible. However, it would be easier for dynamics II, where the linking takes place on the embedded torus rather than the immersed torus, the visualization of which is in itself not easy.

---

## Round 2 · Author Response

Dear Editor,
We would like to resubmit our manuscript to SciPost Physics. We have very much appreciated all the referee's comments and have changed the manuscript accordingly. More background, clarifications, and definitions have been given for the manuscript to be self-contained and more readable for a non-expert. The title has been changed to be more precise, as requested. We hope that this is now suitable for publication.
Best Regards,
the authors.
We would like to resubmit our manuscript to SciPost Physics. We have very much appreciated all the referee's comments and have changed the manuscript accordingly. More background, clarifications, and definitions have been given for the manuscript to be self-contained and more readable for a non-expert. The title has been changed to be more precise, as requested. We hope that this is now suitable for publication.
Best Regards,
the authors.

---

## Round 2 · List of Changes

The title has been changed.
The introduction has been modified. The second sentence now broadly defines "color exchange". The second paragraph mentions the "q-state Potts model" as suggested by a referee. The third paragraph has been partly rewritten to include more background on dimer models (not any longer defined as "two-color" model), and proper definitions. Page 3. Last but one paragraph: more motivation for "dynamics II" is given with some broad references added.
Page 4. Section 2.1. The definitions have been more clearly given and two figures added (fig 1 and 2).
Page 6. Section 2.2. The motivation of "dynamics II" is explained in the first paragraph.
The conclusion has been split into a real conclusion and "perspectives and open problems" (expanded a little).
We have changed the notation for the parity P->R to avoid possible confusions with the polynomial P_5.
The following references were added, to give more background:
C. Castelnovo, P. Pujol, and C. Chamon, Phys. Rev. B \textbf{69}, 104529 (2004),
R. Moessner and K.~S.~Raman, \textit{Highly Frustrated Magnetism}, Eds. C.~Lacroix, P.~Mendels, F.~Mila. Springer Verlag (2010).
V.~I.~Arnol'd, Funct. Anal. Appl., 23:3, 169–177 (1989).
V.~A.~Vassiliev, \textit{Complements of discriminants of smooth maps: topology and applications}, 2-d extended edition, Translations of Math. Monographs, 98, AMS, Providence, RI, 1994, 268 pp.
S.~Fisk, Adv. Math. \textbf{11}, 326 (1973); S.~Fisk, Adv. Math. \textbf{24}, 298 (1977); S. Fisk, Adv. in Math. \textbf{25}, 226 (1977).
O. C\'epas and T. Ziman, Progr. of Theor. Phys. Suppl. 159, 280-291 (2005).
The introduction has been modified. The second sentence now broadly defines "color exchange". The second paragraph mentions the "q-state Potts model" as suggested by a referee. The third paragraph has been partly rewritten to include more background on dimer models (not any longer defined as "two-color" model), and proper definitions. Page 3. Last but one paragraph: more motivation for "dynamics II" is given with some broad references added.
Page 4. Section 2.1. The definitions have been more clearly given and two figures added (fig 1 and 2).
Page 6. Section 2.2. The motivation of "dynamics II" is explained in the first paragraph.
The conclusion has been split into a real conclusion and "perspectives and open problems" (expanded a little).
We have changed the notation for the parity P->R to avoid possible confusions with the polynomial P_5.
The following references were added, to give more background:
C. Castelnovo, P. Pujol, and C. Chamon, Phys. Rev. B \textbf{69}, 104529 (2004),
R. Moessner and K.~S.~Raman, \textit{Highly Frustrated Magnetism}, Eds. C.~Lacroix, P.~Mendels, F.~Mila. Springer Verlag (2010).
V.~I.~Arnol'd, Funct. Anal. Appl., 23:3, 169–177 (1989).
V.~A.~Vassiliev, \textit{Complements of discriminants of smooth maps: topology and applications}, 2-d extended edition, Translations of Math. Monographs, 98, AMS, Providence, RI, 1994, 268 pp.
S.~Fisk, Adv. Math. \textbf{11}, 326 (1973); S.~Fisk, Adv. Math. \textbf{24}, 298 (1977); S. Fisk, Adv. in Math. \textbf{25}, 226 (1977).
O. C\'epas and T. Ziman, Progr. of Theor. Phys. Suppl. 159, 280-291 (2005).

---

## Editorial Decision

published